# Immunology and Immunotherapeutic Approaches for Advanced Renal Cell Carcinoma: A Comprehensive Review

**DOI:** 10.3390/ijms22094452

**Published:** 2021-04-24

**Authors:** Yoon-Soo Hah, Kyo-Chul Koo

**Affiliations:** 1Department of Urology, Catholic University of Daegu School of Medicine, Daegu 42472, Korea; uro.drhah@gmail.com; 2Department of Urology, Gangnam Severance Hospital, Yonsei University College of Medicine, Seoul 06229, Korea

**Keywords:** biomarkers, clinical trials, immune checkpoint inhibitor, immunotherapy, renal cell carcinoma

## Abstract

Renal cell carcinoma (RCC) is a malignant tumor associated with various tumor microenvironments (TMEs). The immune system is activated by the development of cancer and drives T cell anti-tumor response. CD8 T cells are known to improve clinical outcomes and sensitivity to immunotherapy, and play a crucial role against tumors. In contrast, tumor-associated macrophages (TAMs) suppress immunity against malignancy and lead to tumor progression. TAMs are promoted from damaged TMEs and mount proinflammatory responses to pathogens. Initial immunotherapy consists of interferon-α and interleukin-2. However, response to such therapy is unclear in most patients, and it is associated with high levels of toxicity. Immune checkpoint inhibitors (ICIs), which up-regulate immune responses by blocking the programed cell death protein 1 (PD-1) receptor, the ligand of PD-1, or cytotoxic T-lymphocyte-associated protein 4 T cells, have led to a new era of immunotherapy. Furthermore, combination strategies with ICIs have proven effective through several randomized controlled trials. We expect the next generation of immunotherapy to lead to better outcomes based on ongoing trials and inspire new therapeutic strategies.

## 1. Introduction

Renal cell carcinoma (RCC) is the most common type of kidney malignancy, constituting 2–3% of all cancers. This nephron-arising neoplasm consists of heterogenous subgroups according to histologic and molecular subtypes. Clear cell RCC (ccRCC) is an aggressive subtype, constituting 70–80% of all RCCs [1]. Von Hippel-Lindau (VHL) is a crucial component for maintaining the oxygen homeostasis of the cellular environment [2]. The loss of the VHL tumor suppressor drives the hypoxic pathway by hypoxia-inducible factors (HIF) transcription factors. It activates several hypoxia-driven genes, such as vascular endothelial growth factor (VEGF), and subsequently induces angiogenesis and cell growth [3,4]. This VHL mutation course is the main pathway of ccRCC. Modifications of various genes similarly manifest as other types of RCC. Papillary RCC is the second most common subtype of RCC, and is classified into two subtypes: type I, which is mainly associated with MET alterations, and type II, which is associated with the NRF2-antioxidant response component [5]. Chromophobe RCC is associated with mutations of TP53 and PTEN, while translocation RCC is associated with fusions of TFE3 or TFEB genes [6,7].

Decades ago, there were few options for systemic therapy in advanced RCC. Cytokine therapy, represented by interleukin-2 (IL-2) and interferon alfa (IFN-α), showed some benefits in a few advanced patients with RCC, but only proved efficacy in a limited proportion of patients [8]. Moreover, cytokine therapy is associated with a high level of toxicity, which limited its general use. With advances in genomic research by the Cancer Genome Atlas (TCGA), targeted molecular therapeutics, specifically tyrosine kinase inhibitors (TKIs) targeting the VEGF receptor pathway, have now replaced cytokine therapy and are widely used as first- or second-line therapy. The development of the TCGA also led to a better understanding of the mammalian target of the rapamycin (mTOR) pathway that is known to induce cell growth and division in ccRCC [3,9]. Subsequent development and use of mTOR inhibitors have shown similar oncological outcomes to TKIs [10,11,12].

Immunity against malignancy varies depending on several components that make up the tumor microenvironment (TME), and therefore clinical symptoms and the course of treatment differ accordingly. RCC is classified as an immunogenic tumor based on its response to immunotherapy, the incidence of spontaneous regression, and a high level of tumor T cell infiltration [13]. Recent advances in immune checkpoint inhibitor (ICI) therapy up-regulating immune responses by blocking the programed cell death protein 1 (PD-1) receptor, ligand of PD-1 (PD-L1) or cytotoxic T-lymphocyte-associated protein 4 (CTLA-4), and T cells have overtaken cytokine-based regimens and are now key players in the field of immunotherapy [14,15]. Unlike IFN-α and IL-2, for which only a limited range of patients are eligible due to toxicity, ICIs are characterized by superior safety profiles and oncological efficacy. Recently, updated guidelines recommend combining VEGF targeted agents with ICIs depending on patient performance and comorbidity.

Risk stratification systems are essential for selecting the optimal treatment for a specific patient. Unfortunately, there are no predictive biomarkers for RCC, which limits effective strategies for management. Current international guidelines for risk stratification rely on clinical variables to guide prognosis and treatment selection. The Memorial Sloan Kettering Cancer Center (MSKCC) criteria incorporate five prognosticators: low performance status, high level of serum dehydrogenase, high level of serum calcium, low concentration of hemoglobin, and interval less than one year from diagnosis to treatment [16]. The International Metastatic Renal Cell Carcinoma Database Consortium (IMDC) utilizes similar prognosticators to those of MSKCC but includes high levels of neutrophil and platelet counts instead of serum lactate dehydrogenase level [17,18]. Patients without any corresponding prognostic factors are classified into a low-risk group, patients with one or two prognosticators into an intermediate-risk group, and three or more into a poor-risk group.

In this comprehensive review, we will discuss up-to-date evidence on the microenvironments involved in the development of RCC and how treatment strategies targeted at the host immune system are feasible for controlling disease progression. We summarize progress made regarding systemic treatment, from the cytokine therapy era to the treatment utilizing combined ICIs with or without targeted therapies. Lastly, we summarize ongoing trials involving immunotherapies that will change the landscape of future systemic therapies targeted at advanced RCC.

## 2. Tumor Microenvironment in Renal Cell Carcinoma

Chromosome 3p loss is the first genetic event characterizing sporadic ccRCC, followed by VHL mutation [19]. VHL negatively regulates HIF 1/2α, which reduce oxygen demand in the cellular environment by increasing glycolytic flux and reducing oxidative phosphorylation. This pathway induces oxygen supply by hyper-vascularization. Based on metabolic pathway analysis by RNA sequencing, ccRCC is known to possess high levels of metabolites during glycolysis and to reduce levels of metabolites associated with oxidative phosphorylation [20]. Hyper-vascularity and the immune system are not independent, and treatment targeting the VEGF receptor promotes the immune pathway by modifying the aberrant blood supply [21].

TME are complicated, containing transformed cells and immune infiltrates. Tumor-infiltrating cells promote or inhibit cancer activity according to the type of cancer. The immune system is activated by cancer development and drives T cell anti-tumor response by suppressing tumor cells directly, modulating various anti-tumor responses, facilitating the emerging memorial system, and preparing specificity for tumor-derived proteins [8]. T cell activation, according to immunotherapy response, is a core component in the prognosis of ccRCC. CD8 T cells play a crucial role in combating malignant tumors and are associated with favorable clinical outcomes and response to immunotherapy [22,23,24,25,26]. Presentation of major histocompatibility class I (MHC-I) molecules on cancer cells helps T cell receptors recognize antigens. This pathway activates CD8 T cells, which subsequently activates antigen-specific immune response, and directly removes antigen-bearing cells [27]. The antigen-presenting machinery (APM) promoted by activated CD8 T cells is a component that interlinks antigens and MHC-I. Upregulation of APM genes refers to the increased production of antigen-presentation and the number of T cells. CcRCC is characterized by the highest T cell infiltration and immune infiltration when compared to other malignancies. The immunogenicity of ccRCC is related to MHC-I and APM gene expression, which may potentially serve as indicators of response to PD-1 inhibitors. The promotion of APM expression is a unique feature of ccRCC [28]. In contrast, Th2 and regulatory T cells are negatively associated with prognosis. An abundant environment with Th2 and regulatory T cells suppresses the immune response and is associated with the tumor mutation load [28].

Macrophages are phagocytic innate immune cells that regulate responses to tissue damage. Macrophages are promoted based on consecutive signals from the damaged microenvironment and mount proinflammatory responses to pathogens [29]. Macrophages are abundantly observed in growing cancer cells and mediate lymphocyte trapping according to interactions with CD8 T cells in tumor stroma. Cytokines and chemokines expressed by tumor-associated macrophages (TAMs) suppress immunity against malignancy and lead to tumor progression [30]. In an in vivo study, the efficiency of T cells increased when TAMs were depleted by pexidartinib, a small molecule tyrosine kinase inhibitor that acts against colony-stimulating factor 1. The depletion of TAMs not only increased the number of tumor-infiltrating CD8 T cells but also improved their migration and ability to reach cancer cells [31,32,33].

Tertiary lymphoid structures (TLS) are a lymphoid environment usually associated with reactions to infection or inflammation [34]. TLS neogenesis is induced by chronic bacterial or viral infection or by chronic inflammatory diseases such as multiple sclerosis, Sjögren’s syndrome, or allograft rejection [35,36,37,38,39]. TME is similar to TLS and includes several components associated with the immune system and T cell activation. Mature dendritic cells (DCs), which are associated with activated CD8 T cells within the TLS, are associated with favorable survival outcomes in ccRCC. On the other hand, DCs outside of TLS are associated with poor survival outcomes in response to dysfunctional CD8 T cells [40,41].

The wide variety of clinical features and outcomes of immunotherapy in patients with ccRCC are due to the heterogeneity of TME. A study that utilized mass cytometry confirmed the subsets of T cells and TAMs, the critical components of TME [42]. In a study comparing 73 patients with ccRCC and five healthy controls, 20 T cell phenotypes and 17 TAMs phenotypes were identified. With ongoing research on the mechanisms of treatment failure, TME heterogeneity is being perceived as a key factor.

## 3. Immunotherapy in the Early Era

Late recurrence after partial or radical nephrectomy, long-term stabilization of disease without systemic treatment, and, in rare cases, spontaneous regression suggest that the mechanisms of the host immune system are keystones of controlling tumor growth or suppression [43,44,45,46].

IFN-α has been the primary agent used in the early immunotherapy era. The overall response rate to IFN-α has been reported to be low as 12% [43]. Patients with visceral metastatic RCC, particularly lung RCC, or with prior nephrectomy showed more favorable survival outcomes [47,48]. However, maintenance of response was restricted to less than two years [43].

Patients treated with high-dose IL-2 exhibit 4% complete response (CR), 8% partial response, and 23 months of response duration. Moreover, high-dose IL-2 was related to severe cardiovascular toxicity due to increased vascular permeability, with treatment-related death occurring in 4% of patients [49,50]. Although oncological outcomes with IFN-α or IL-2 are often dramatic in selected patients, most patients experienced no apparent survival benefit. Hence, these agents are not considered as first- or second-line therapy unless the patient has excellent performance status with normal organ function [8].

Combination therapy consisting of IFN-α and bevacizumab was studied in several trials. Patients with metastatic RCC without previous treatment received IFN-α plus bevacizumab or IFN-α with placebo. Progression-free survival (PFS) was superior in the combination treatment arm. However, no significant improvements were observed in overall survival (OS) during the study period [51,52,53,54].

## 4. Immune-Associated Novel Prognosticators of Renal Cell Carcinoma

Recent genomic studies of RCC have developed an understanding of tumor heterogeneity [3,55,56]. Much progress has been made in revealing the relationships between the immune system and tumors, but immunotherapy responses differ in each case. Even within a specific type of cancer, a subset of patients will show strong immune infiltration, while others show little or no response to immunotherapy. This heterogeneity suggests that pathological classification alone is insufficient to predict treatment effect and prognosis. A more detailed sub-classification system is warranted [56,57].

Long non-coding RNAs (lncRNAs) are predictors that have become an important focus of debate in recent years [58,59]. LncRNAs are non-protein coding RNAs longer than 200 nucleotides [60,61] that are involved in tumor development and suppression by regulating the immune system [62,63,64,65]. LncRNAs are more tumor-specific than protein-coding RNAs, inducing up-regulated RCC carcinogenesis, promoting progression and metastasis with a positive-feedback loop [66].

Khadirnaikar et al. found 143 immune-associated lncRNAs genes related to RCC by examining 2378 genes in TCGA RNA sequence data [67]. They divided gene samples into three groups (C1, C2, and C3) by K-means consensus clustering according to the expression levels of immune lncRNAs. In a classification according to immune cluster, C3 showed a significantly higher grade of tumor and metastasis and thus poorer prognosis than other groups. These immune clusters yielded better survival prediction rates than those using miRNA or mRNA. The analysis showed increased CD8 T cells and decreased DCs in the C3 cluster when compared to other groups. Patients with C3 showed higher CD8 T cell infiltration; however, they had a worse prognosis due to lack of DC, which plays a role in T cell activation. On the other hand, in C1 and C2, naïve B cell and neutrophil infiltration associated with a better prognosis were higher than C3 [68,69]. Immune infiltration is different depending on the expression level of lncRNAs, which may explain the differences in prognosis.

Molecular subsets of RCC have been redefined according to differential clinical responses to angiogenesis blockade with or without ICI [70,71]. Motzer et al. performed transcriptomic analysis of advanced RCC tumor samples, which revealed seven subsets with distinct angiogenesis, immune, cell-cycle, metabolism, and stromal programs. Clinical benefits with sunitinib and atezolizumab plus bevacizumab were observed in patients with high angiogenesis, while atezolizumab plus bevacizumab was beneficial in patients with high T-effector and/or cell-cycle transcription [57]. Overall, the results implied that RCC may be molecularly classified to stratify patients for the optimal oncological outcome.

## 5. Immune Checkpoint Inhibitors for Renal Cell Carcinoma

Blockades of immune checkpoint components such as PD-1/PD-L1 and CTLA-4 have shown considerable oncological benefit and have shifted treatment strategies targeted at RCC. Clinical trials involving ICIs are summarized in Table 1.

CheckMate 025 was a phase III, open-label, randomized study that compared nivolumab with everolimus. A total of 821 patients with advanced ccRCC who had received previous anti-angiogenic therapy were randomly allocated to receive nivolumab (3 mg/kg) every two weeks or everolimus (10 mg) daily [72]. The median OS was favorable for nivolumab when compared to everolimus (25.0 months vs. 19.6 months; HR 0.73; 98.5% confidence interval [CI] 0.57–0.93; *p* = 0.002). The objective response rate (ORR) was also superior in the nivolumab arm (25% vs. 5%; 95% CI 3.68–9.72; *p* < 0.001). However, PFS was comparable in both treatment arms (4.6 months vs. 4.4 months; HR 0.88; 95% CI 0.75–1.03; *p* = 0.11). Superior quality-of-life (QoL) was observed in patients treated with nivolumab, with fewer treatment-related adverse events (TRAEs) of grade 3 or 4 (19% vs. 37%) [73].

Sunitinib and pazopanib are common first-line agents used for patients with advanced RCC. The recent development of combination treatments with ICIs is changing treatment paradigms, especially for intermediate-risk or poor-risk patients. CheckMate 214 was a phase III study that compared nivolumab and ipilimumab with sunitinib for patients with treatment-naïve advanced ccRCC [74]. A total of 1096 patients were allocated at a 1:1 ratio to nivolumab (3 mg/kg) every two weeks and ipilimumab (1 mg/kg) every three weeks or sunitinib (50 mg) daily for four weeks (6-week cycle). In patients with intermediate-risk or poor-risk group by the IMDC criteria, the median OS was not reached in the combination group, while 26 months was achieved in the sunitinib group (HR 0.63; *p* < 0.001). The ORR was superior in the combination group compared to the sunitinib group a 42% versus 27% (*p* < 0.001). CR rates were also favorable in the combination group compared to the sunitinib group (9% vs. 1%; *p* < 0.001). Median PFS was improved with combination therapy (11.6 months vs. 8.4 months; HR 0.82; *p* = 0.03); however, it did not satisfy the prespecified statistical threshold (alpha level = 0.009). Grade 3 or 4 TRAEs were observed in 46% of the patients in the nivolumab and ipilimumab combination group, while 63% of the patients in the sunitinib group. The patients were administered FKSI-19 questionnaires to access health-related QoL, which revealed that the combination arm experienced more significant improvement from baseline than the sunitinib arm (*p* < 0.001). ORR was lower in the combination arm than the sunitinib arm (29% vs. 52%; *p* < 0.001). Median PFS was also inferior in the combination arm than in the sunitinib arm (15.3 months vs. 25.1 months; HR for progressive disease or death, 2.18; 99.1% CI, 1.29–3.68; *p* < 0.001).

IMmotion151 was a phase III, open-label, randomized study that compared atezolizumab plus bevacizumab with sunitinib for chemotherapy-naïve advanced RCC patients with clear cell or sarcomatoid pathology [75,76]. A total of 915 patients were randomly allocated at a 1:1 ratio to atezolizumab (1200 mg) and bevacizumab (15 mg/kg) every three weeks or sunitinib (50 mg) daily for four weeks (6-week cycle). Overall, 40% of the patients exhibited PD-L1 expression, with more than 1% in tumor-infiltrating immune cells. In PD-L1 positive patients, the median PFS was superior in the combination arm compared to the sunitinib arm (11.2 months vs. 7.7 months; HR 0.74; 95% CI 0.57–0.96; *p* = 0.022). The intention-to-treat (ITT) cohort exhibited similar favorable results in the combination arm (11.2 months vs. 8.4 months; HR 0.83; 95% CI 0.70–0.97; *p* = 0.022). However, median OS was comparable in both PD-L1 positive (HR 0.84; 95% CI 0.62–1.15; *p* = 0.286) and in ITT patients (HR 0.93; 95% CI 0.76–1.14; *p* = 0.475) in the second interim analysis. In subgroup analyses, PD-L1 positive patients who were administered atezolizumab plus bevacizumab exhibited superior PFS regardless of MSKCC and IMDC risk classification criteria. Previous nephrectomy and the absences of liver metastasis and sarcomatoid histology were factors associated with favorable PFS in the combination arm. ORR was superior with atezolizumab plus bevacizumab (43%) than with sunitinib (35%) in PD-L1 positive patients. The rate of CR was comparable between the combination versus the sunitinib arms (9% vs. 4%). Grade 3 or 4 TRAEs were noted in 40% of patients in the combination arm and 54% of patients in the sunitinib arm.

**Table 1 ijms-22-04452-t001:** Phase III clinical trials investigating combination immune checkpoint inhibitor therapies for advanced renal cell carcinoma.

Trial	Agents	Clinical Setting	OS (Months)	PFS (Months)	ORR (%)	TRAEs (%) *
CheckMate 025 [72]	Nivolumab vs. Everolimus	Second-line	25.0 vs. 19.6	4.6 vs. 4.4	25.0 vs. 5.0	19.0 vs. 37.0
			(*p* = 0.002)	(*p* = 0.11)	(*p* < 0.001)	
CheckMate 214 [74]	Nivolumab + Ipilimumab vs. Sunitinib	First-line Intermediate- or poor-risk	Not reached vs. 26.0	11.6 vs. 8.4	42.0 vs. 27.0	46.0 vs. 63.0
			(*p* < 0.001)	(*p* = 0.03) ⁑	(*p* < 0.001)	
IMmotion151 [75]	Atezolizumab + Bevacizumab vs. Sunitinib	First-line PD-L1 +, ITT	34.0 vs. 32.7 ^†^	11.2 vs. 7.7	43.0 vs. 35.0	40.0 vs. 54.0
			(*p* = 0.286)	(*p* = 0.0217)		
JAVELIN Renal 101 [77]	Avelumab + Axitinib vs. Sunitinib	First-line PD-L1 +	Patients continued to be followed	13.8 vs. 7.2	55.2 vs. 25.5 ^‡^	71.2 vs. 71.5
				(*p* < 0.001)		
KEYNOTE-426 [78]	Pembrolizumab + Axitinib vs. Sunitinib	First-line	Not reached in both groups	15.1 vs. 11.1	59.3 vs. 35.7	62.9 vs. 58.1
				(*p* < 0.001)	(*p* < 0.001)	

* Treatment-related adverse event grade 3 or 4. ⁑ Not significant per the prespecified alpha level 0.009 threshold. ^†^ Not estimated at the second interim analysis. ^‡^ The stratified odds ratio 3.73. ITT, intention-to-treatment; ORR, objective response rate; OS, overall survival; PD-L1, programed death-ligand 1; PFS, progression-free survival; TRAEs, treatment-related adverse events.

JAVELIN Renal 101 also involved a combination of ICIs as the first-line of therapy [77]. This phase III trial involved 886 patients with treatment-naïve advanced RCC and randomized patients at a 1:1 ratio to compare avelumab (10 mg/kg) every two weeks plus axitinib (5 mg) twice daily with sunitinib (50 mg) once daily for four weeks (6-week cycle). In the PD-L1 positive patients, the median PFS was favorable for avelumab plus axitinib compared to sunitinib (13.8 months vs. 7.2 months; HR 0.61; 95% CI 0.47–0.79; *p* < 0.001). The ORR in the avelumab and axitinib group was higher than that of the sunitinib group in both the PD-L1 positive group and the overall group (55.2% vs. 25.5% and 51.4% vs. 25.7%, respectively). The rate of CR was also higher in the combination group in both the PD-L1 positive group and in the overall group (12% vs. 6% and 15% vs. 8%, respectively). TRAEs appeared in 99.5% of patients who received avelumab plus axitinib, but in 99.3% of patients who received sunitinib. More patients needed subsequent therapy after sunitinib compared to combination therapy (39.2% vs. 20.8%).

KEYNOTE-426 was a phase III study that compared pembrolizumab plus axitinib with sunitinib. A total of 861 patients with chemotherapy-naïve, advanced RCC were randomly allocated at a 1:1 ratio to receive pembrolizumab (200 mg) every three weeks with axitinib (5 mg) twice daily or sunitinib (50 mg) daily for four weeks (6-week cycle) [78]. The estimated survival rate at 12 months was 89.9% in the pembrolizumab plus axitinib arm and 78.3% in the sunitinib arm. The median survival was not reached in both arms; however, the possibility of death was significantly lower in the combination therapy arm (HR 0.53; 95% CI 0.38–0.74; *p* < 0.001). The median PFS was four months longer in the pembrolizumab and axitinib arm than in the sunitinib arm (15.1 months vs. 11.1 months; HR 0.69; 95% CI 0.57–0.84; *p* < 0.001). The survival advantages of pembrolizumab plus axitinib were observed regardless of IMDC risk classification and PD-L1 expression. The ORR and CR were superior in the combination arm compared to the sunitinib arm (59.3% vs. 35.7%; *p* < 0.001 and 5.8% vs. 1.9%, respectively). Subsequent chemotherapy was needed in 50.0% of patients in the pembrolizumab plus axitinib arm and in 60.7% of the patients in the sunitinib arm. TRAEs were observed in 96.3% of the patients in the pembrolizumab-axitinib arm and 97.6% of the patients in the sunitinib arm. Notably, the rate of grade 3 or higher TRAEs was higher in the pembrolizumab-axitinib arm than the sunitinib arm (62.9% vs. 58.1%). An extended study with a median follow-up of 30.6 months also showed benefits in the pembrolizumab plus axitinib combination arm [71]. The OS was not reached in the pembrolizumab-axitinib arm, while 35.7 months was observed in the sunitinib arm within the ITT population (HR 0.68; 95% CI 0.55–0.85; *p* < 0.001). In this extended exploratory analysis, patients with favorable-risk based on the IMDC criteria showed no difference in OS (HR 1.06; 95% CI 0.60–1.86; *p* = 0.58), while patients with intermediate-risk or poor-risk showed significant benefit (HR 0.63; 95% CI 0.50–0.81; *p* < 0.001).

## 6. Ongoing Trials Involving Immune Checkpoint Inhibitors

Several phase III clinical trials involving ICIs are ongoing and are expecting results [79]. A summary of these ongoing trials is presented in Table 2.

KEYNOTE-679/ECHO-302 is a phase III, open-label, randomized controlled trial comparing pembrolizumab plus epacadostat with standard TKI treatment such as sunitinib or pazopanib in patients with treatment-naïve, locally advanced or metastatic ccRCC (NCT03260894) [80]. Patients of the combination therapy group receive pembrolizumab (200 mg) intravenously every three weeks and epacadostat (100 mg) orally twice daily. Standard-of-care patients receive sunitinib (50 mg) once daily for four weeks (6-week cycle) or pazopanib 800 mg once daily. The primary endpoint is ORR, while the secondary endpoints are safety and tolerability.

CLEAR is a phase III randomized study comparing lenvatinib in combination with everolimus or pembrolizumab versus sunitinib alone in the first-line setting of advanced RCC (NCT02811861) [81]. Patients who receive lenvatinib (18 mg) daily plus everolimus (5 mg) daily or lenvatinib (20 mg) daily plus pembrolizumab (200 mg) every three weeks are compared with patients who receive sunitinib (50 mg) once daily for four weeks (6-week cycle). The primary endpoint is PFS, and the secondary endpoints are ORR, OS, TRAEs, health-related QoL, and PFS until the next-line of therapy.

CheckMate 9ER is a phase III, open-label, randomized trial of nivolumab combined with cabozantinib versus sunitinib in patients with previously untreated advanced or metastatic RCC (NCT03141177) [82]. The primary endpoint is PFS, and the secondary endpoints are OS, ORR, and TRAEs.

COSMIC-313 is a phase III, double-blind, randomized trial comparing cabozantinib in combination with nivolumab and ipilimumab (four doses) versus nivolumab and ipilimumab (four doses) in patients with treatment-naïve, advanced or metastatic ccRCC of intermediate-risk or poor-risk (NCT03937219) [83]. The primary endpoint is PFS, and the secondary is OS.

PDIGREE is a phase III, open-label, randomized trial comparing nivolumab and ipilimumab followed by nivolumab versus cabozantinib with nivolumab in patients with untreated metastatic RCC (NCT03793166) [84]. Patients receive nivolumab and ipilimumab intravenously every three weeks for up to four cycles. Patients with progressive disease receive cabozantinib daily until further disease progression or unacceptable toxicity. Patients with CR continue nivolumab intravenously every four weeks. Patients with a non-CR and non-progressive disease receive nivolumab intravenously every four weeks or nivolumab every four weeks plus cabozantinib daily in the absence of disease progression or unacceptable toxicity. The primary endpoint is OS, and the secondary endpoints are PFS, CR, ORR, TRAEs.

CONTACT-03 is a phase III, open-label, randomized trial to investigate the efficacy and safety of atezolizumab plus cabozantinib versus cabozantinib monotherapy in patients with inoperable, locally advanced, or metastatic RCC who exhibit radiographic tumor progression during or after ICI treatment (NCT04338269) [85]. Patients with disease progression after treatment of atezolizumab, avelumab, pembrolizumab, or nivolumab receive atezolizumab (1200 mg) every three weeks with cabozantinib (60 mg) orally once daily or cabozantinib alone. The primary endpoints are PFS and OS. Ongoing trials associated with surgical treatments are summarized in Table 2 [86,87,88,89].

## 7. Summary and Future Directions

Advancements in ICIs have led to improved therapeutic efficacy and safety for various types of tumors, including advanced RCC. The immune mechanisms underlying the development and progression of RCC make ICIs the most valuable potential systemic therapy in RCC management.

Early cytokine immunotherapy played an important role in the management of advanced RCC, but its high toxicity profile and low response rate limited its widespread use. TKIs targeting the VEGF receptor pathway have made significant advancements in TKIs without adverse events. Newly developed ICIs and their combined treatments have shown favorable results in terms of oncological outcomes and safety profiles and are currently recommended as first-line therapy.

Cytokines and chemokines expressed by TAMs suppress anti-tumor immune mechanisms, leading to tumor progression [30,90,91]. Furthermore, TAMs are known strongly associated with resistance to TKIs and ICIs. Specific pathways regulating the recruitment, polarization, and metabolism of TAMs have been identified in preclinical studies [92].

Understanding TME is an important step in understanding immune mechanisms involved in RCC development. TKI inhibits the process in which mutations of the VHL gene induces HIF to accelerate VEGF for neovasculation and tumor development. Novel approaches are underway to develop biomarkers associated with TAM and to integrate novel radiomic modalities. Accumulation of mannosylated liposome containing fluorescent dye in TAMs has been demonstrated in a mouse model of lung carcinoma [93]. This indicates that mannose-coated liposomes combined with therapeutic agents could be delivered to TME. A pilot study that quantified TAM using ferumoxytol-enhanced MRI illustrated the possibility that TME could be accessed with modified MR technology [94]. New approaches and applications of functional and structural imaging for RCC are being investigated, and are expected to be useful decision-making tools in the near future [95,96,97].

Recent trials have shown that combined ICI therapies are oncologically superior to single-agent targeted therapies in terms of OS and PFS outcomes as well as TRAEs profiles. Future studies are warranted to elucidate the optimal combination and sequencing of these agents for maximal survival benefit. Further research on novel diagnostic modalities remains to be performed.

## Figures and Tables

**Table 2 ijms-22-04452-t002:** Ongoing phase III clinical trials investigating first-line therapies for advanced renal cell carcinoma.

Trial	Identifier	Comparing Agents	Primary Endpoint
KEYNOTE-679/ECHO-302 [80]	NCT03260894	Pembrolizumab + Epacadostat vs. Sunitinib or Pazopanib	ORR
CLEAR [81]	NCT02811861	Lenvatinib + (Everolimus or Pembrolizumab) vs. Sunitinib	PFS
CheckMate 9ER [82]	NCT03141177	Nivolumab + Cabozantinib ± Ipilimumab vs. Sunitinib	PFS
COSMIC-313 [83]	NCT03937219	Cabozantinib + Nivolumab + Ipilimumab vs. Nivolumab + Ipilimumab	PFS
PDIGREE [84]	NCT03793166	Nivolumab + Ipilimumab → Nivolumab + Cabozantinib vs. Nivolumab	OS
CONTACT-03 [85]	NCT04338269	(Atezolizumab or Avelumab or Nivolumab or Pembrolizumab) → PD → Atezolizumab + Cabozantinib vs. Cabozantinib	OS, PFS
CheckMate 914 [86]	NCT03138512	Radial or partial nephrectomy → Nivolumab ± Ipilimumab vs. Placebo	DFS
NORDIC-SUN [87]	NCT03977571	Cytoreductive nephrectomy + Nivolumab + Ipilimumab → Nivolumab vs. No surgery + Nivolumab + Ipilimumab → Nivolumab	OS
PROSPER RCC [88]	NCT03055013	Radical or partial nephrectomy + perioperative Nivolumabvs. Radial or partial nephrectomy only	PFS
RAMPART [89]	NCT03288532	Radical or partial nephrectomy → (Active monitoring vs. Durvalumab vs. Durvalumab + Tremelimumab)	DFS, OS

DFS; disease-free survival, ORR, objective response rate; PD, progressive disease; PFS, progression-free survival.

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
