# Peer review of "Immunology and Immunotherapeutic Approaches for Advanced Renal Cell Carcinoma: A Comprehensive Review"

_ijms, 2021, doi:10.3390/ijms22094452_

Round 1
Reviewer 1 Report
Review comment
(Manuscript: ijms-1154725)
The authors have elegantly summarized tumor-associated immunology involved in renal cell carcinoma and current advancements in immunotherapy. The manuscript has been well-written, but there are also several points that need to be addressed for further improvement.
[Major]
1. The authors introduce ongoing trials involving immune checkpoint inhibitors for renal cell carcinoma. There are more unmentioned phase III trials, such as CheckMate 914 and NORDIC-SUN, which involves patients receiving cytoreductive nephrectomy. There is another that investigates nivolumab for patients with localized disease undergoing nephrectomy. Any particular reason why these trials were not described? I would recommend at least a brief discussion on these trials.
2. Section 4 summarizes immunotherapeutic strategies in the early era, mostly out-of-use in the contemporary era. The authors may cut down the discussed contents for the brevity of the manuscript.
[Minor]
1. The article contains several abbreviations. Unnecessary abbreviations could be sorted out according to the journal’s guidelines.
2. Third paragraph, Section 2. ‘Tumor Microenvironment in Renal Cell Carcinoma.’ Please check for any errors in terminologies that were used.
Author Response
The authors have elegantly summarized tumor-associated immunology involved in renal cell carcinoma and current advancements in immunotherapy. The manuscript has been well-written, but there are also several points that need to be addressed for further improvement.
REPLY: We are very much thankful to the reviewer for the thorough review. We agree with all the specific comments raised and have revised our paper in light of the useful suggestions. Our responses to the specific comments/suggestions/queries given are provided below.
[Major]
1. The authors introduce ongoing trials involving immune checkpoint inhibitors for renal cell carcinoma. There are more unmentioned phase III trials, such as CheckMate 914 and NORDIC-SUN, which involves patients receiving cytoreductive nephrectomy. There is another that investigates nivolumab for patients with localized disease undergoing nephrectomy. Any particular reason why these trials were not described? I would recommend at least a brief discussion on these trials.
REPLY: We thank the reviewer for this suggestion. We had not mentioned these trials in our first draft for the brevity of the manuscript. There exist many more ongoing trials involving immune checkpoint inhibitors for renal cell carcinoma. According to your suggestion, we added information on CheckMate 914 (NCT03138512), NORDIC-SUN (NCT03977571), PROSPER RCC (NCT03055013), and RAMPART (NCT03288532) to Table 2. However, we did not add detailed descriptions of these trials in the text, since the results of the trials have not yet been published.
2. Section 4 summarizes immunotherapeutic strategies in the early era, mostly out-of-use in the contemporary era. The authors may cut down the discussed contents for the brevity of the manuscript.
REPLY: We agree that immunotherapeutic strategies in the early era are now out-of-use and does not deserve such a detailed description. We removed the unnecessary sentences to make our manuscript more concise.
[Minor]
1. The article contains several abbreviations. Unnecessary abbreviations could be sorted out according to the journal’s guidelines.
REPLY: Thank you for pointing this out. We avoided using abbreviations that were used less than three times.
2. Third paragraph, Section 2. ‘Tumor Microenvironment in Renal Cell Carcinoma.’ Please check for any errors in terminologies that were used.
REPLY: We had made a typo in “hyper-vasculation.” This has now been corrected.
Reviewer 2 Report
Yoon Soo Hah and Kyo Chul Koo have performed a review focused on immunology and therapies in RCC. Overall is a reasonable review, however there are important issues.
First, the immunotherapy section is totally outdated. Second, there are relevant papers about immunotherapy and RCC not mention (ie. Motzer, Cancer Cell 2020) and third the flow is improvable
Minor comments.
RCC is one of the most immune-related tumors in pan-cancer comparisons... vague sentence, please specify differences.
there are no prognostic biomarkers for RCC. This sentence is not true. There are many prognostic biomakers. There are no predicitve biomarkers.
Author Response
Yoon Soo Hah and Kyo Chul Koo have performed a review focused on immunology and therapies in RCC. Overall is a reasonable review, however there are important issues.
REPLY: We are very much thankful to the reviewer for the thorough review. We agree with all the specific comments raised and have revised our paper in light of the useful suggestions. Our responses to the specific comments/suggestions/queries given are provided below.
First, the immunotherapy section is totally outdated. Second, there are relevant papers about immunotherapy and RCC not mention (ie. Motzer, Cancer Cell 2020) and third the flow is improvable
REPLY: Thank you for the comment. We agree that the “immunotherapeutic strategies in the early era” section describes mostly out-of-use strategies in the contemporary era. Since immunotherapeutic strategies of the early era do not deserve such detailed descriptions, we removed unnecessary contents for the brevity of the manuscript.
Thank you for providing us with the information on the molecular subsets of RCC (Motzer et al. Cancer Cell 2020; 38: 803). Molecular subsets of RCC have been redefined according to the differential clinical responses to angiogenesis blockade with or without ICI. This study reports results from transcriptomic analysis of advanced RCC tumor samples, which revealed seven subsets with distinct angiogenesis, immune, cell-cycle, metabolism, and stromal programs. Clinical benefits with sunitinib and atezolizumab plus bevacizumab were observed in patients with high angiogenesis, while atezolizumab plus bevacizumab was beneficial in patients with high T-effector and/or cell-cycle transcription. Overall, the results implied that RCC may be molecularly classified to stratify patients for the optimal oncological outcome. We added this information to Section 4, and also modified the subtitle of the section accordingly.
We also agree that the flow of the manuscript can be improved by allocating the “4. Immunotherapy in the Early Era” section prior to “3. Immune Associated Novel Prognosticators of Renal Cell Carcinoma” section. We edited the format accordingly.
RCC is one of the most immune-related tumors in pan-cancer comparisons... vague sentence, please specify differences.
REPLY: We agree that this sentence was vague. We edited this sentence as follows: “RCC is classified as an immunogenic tumor based on its response to immunotherapy, the incidence of spontaneous regression, and a high level of tumor T cell infiltration.”
There are no prognostic biomarkers for RCC. This sentence is not true. There are many prognostic biomakers. There are no predicitve biomarkers.
REPLY: We agree that “predictive biomarkers“ is the correct terminology. We edited this terminology accordingly.
Reviewer 3 Report
The Review entitled ‘Immunology and Immunotherapeutic Approaches for Advanced Renal Cell Carcinoma: A Comprehensive Review’ by Yoon et.al provides an insight into the microenvironments involved in the development of Renal cell carcinoma and treatment strategies targeted at the host immune system controlling disease progression.
In the introduction authors have briefly discussed the major genes and pathways involved in the development of RCC and from the cytokine therapy era to the contemporary treatment utilizing combined immune checkpoint inhibitors
The authors have discussed the tumor microenvironment in RCC, immune associated long non-coding RNAs as novel prognostic markers of RCC and ongoing trials involving immune checkpoint inhibitors which is very interesting and most important part of this review.
Overall, the manuscript is very well written and is within the scope of the Journal. The authors have reviewed and included data from latest research articles and presented in tabular form which will give a better understanding to the readers.
The review article may be considered for publication in its present form.
Author Response
The Review entitled ‘Immunology and Immunotherapeutic Approaches for Advanced Renal Cell Carcinoma: A Comprehensive Review’ by Yoon et.al provides an insight into the microenvironments involved in the development of Renal cell carcinoma and treatment strategies targeted at the host immune system controlling disease progression. In the introduction authors have briefly discussed the major genes and pathways involved in the development of RCC and from the cytokine therapy era to the contemporary treatment utilizing combined immune checkpoint inhibitors. The authors have discussed the tumor microenvironment in RCC, immune associated long non-coding RNAs as novel prognostic markers of RCC and ongoing trials involving immune checkpoint inhibitors which is very interesting and most important part of this review.
Overall, the manuscript is very well written and is within the scope of the Journal. The authors have reviewed and included data from latest research articles and presented in tabular form which will give a better understanding to the readers.
The review article may be considered for publication in its present form.
REPLY: We are very much thankful to the reviewer for the thorough review.